# The Distribution of Aircraft Icing Accretion in China—Preliminary Study

**Jinhu Wang [1,2,3,4,*]**, **Binze Xie [1,*]** and **Jiahan Cai [1]**

1   Collaborative Innovation Center on Forecast and Evaluation of Meteorological Disasters, Key Laboratory for Aerosol-Cloud-Precipitation of China Meteorological Administration, Nanjing University of Information Science and Technology (NUIST), Nanjing 210044, China; 20181204001@nuist.edu.cn
2   Key Laboratory of Middle Atmosphere and Global Environment Observation, Institute of Atmospheric Physics, Chinese Academy of Sciences, Beijing 100029, China
3   National Demonstration Center for Experimental Atmospheric Science and Environmental Meteorology Education, Nanjing University of Information Science and Technology, Nanjing 210044, China
4   Nanjing Xinda Institute of Safety and Emergency Management, Nanjing 210044, China
*   Correspondence: goldtigerwang@nuist.edu.cn (J.W.); 20181204022@nuist.edu.cn (B.X.); Tel.: +86-138-1451-2847 (J.W.)

**Abstract:** The icing environment is an important threat to aircraft flight safety. In this work, the icing index is calculated using linear interpolation and based on temperature and relative humidity (RH) curves obtained from radiosonde observations in China. The results show that: (1) there are obvious differences in icing index distribution in parameter over various climatic regions of China. The differences are reflected in duration, main altitude, and ice intensity. The reason for the differences is related to the temperature and humidity environment. (2) Before and after the summer rainfall process, there are obvious changes in the ice accretion index in the 4–6 km altitude area of Northeast China, and the areas with serious ice accretion are coincident with areas with large rainfall estimates. (3) In the process of snowfall in winter, the ground snow has an impact on the ice accumulation index in the east of China. When it is snowing, ice accumulation in low altitudes is serious. The results of this study offer a theoretical basis for prediction and early warning of aircraft icing.

**Keywords:** aviation flight safety; icing index; interpolation; China's icing climatic region; rainfall; snowfall

## 1. Introduction

With the development of the civil aviation industry, a large number of aviation accidents have been caused by icing [1–3]. Aircraft icing has the following environmental characteristics: (1) the sky is mainly cloudy, and stratiform cloud easily induce icing or even serious icing [4]; (2) the main phase state of the condensate is liquid, and when an aircraft passes over an area containing super-cooled water droplets, the aircraft is prone to icing [5–13]; (3) a low temperature and high humidity environment is conducive to aircraft icing, with an external temperature generally between −15 °C and −3 °C, and the higher the humidity, the easier it is for. Approximately 75% of aircraft icing occurs when RH is greater than 70% [8,14,15]. Previous studies have shown that icing might change the shape of the wing surface and even affect its aerodynamic characteristics, leading to flight accidents [16–18].

Using the median volume diameter (MVD) to describe the distribution of liquid water content (LWC) at the cloud droplet scale can further describe the icing environment [19]. LWC can be obtained as a function of collection efficiency, MVD, integrated total droplet number concentration, true airspeed, and precipitation rate [20]. According to the "Part 25—Airworthiness Standards: Transport Category Airplanes" specified by the Federal Aviation Administration (FAA), appendix C specifies atmospheric

icing conditions and airframe ice accretions, and appendix O specifies super-cooled large droplets (SLD) icing conditions. In appendix C, the maximum continuous or intermittent intensity of atmospheric icing conditions are defined by the variables of the cloud LWC, the mean effective diameter of the cloud droplets, the ambient air temperature. In appendix O, SLD icing conditions are defined by the parameters of altitude, vertical and horizontal extent, temperature, liquid water content, and water mass distribution as a function of drop diameter distribution [21]. Civil Aviation Administration of China (CAAC) has made similar regulations. Review by Cao et al. [22] indicated that common causes of aircraft freezing include: the aircraft encounters clouds with super-cooled water droplets during flight, the surface of the aircraft has been contaminated before takeoff, or the aircraft encounters high concentrations of ice crystals during flight. The World Area Forecast Centers (WAFC) calculated an icing potential based on RH, where the cloud is present and the temperature is between 0 °C and −20 °C, otherwise icing potential is set to 0. The forecasts from each WAFC are combined to produce harmonized forecasts available to the global aviation community [23]. Although LWC and droplets MVD are directly related to icing in flight, however, the inversion of cloud microphysical characteristics is a complex problem, and the results of the algorithm are not easy to verify [20,24–30]. As far as current research is concerned, the research on icing numerical weather prediction for aviation operations is still immature [31]. The area with high humidity is more likely to appear super-cooled water, and RH is easy to be measured by remote sensing. Therefore, the ice accretion index is calculated according to temperature and RH, which represents the possibility of icing.

Many researchers have proposed a variety of classic ice accretion prediction algorithms [6,32–34]. Seongmun et al. [35] put forward the ice detection model based on machine learning. Merino et al. [23] used a C-212 to measure the cloud microphysical characteristics of 37 areas containing super-cooled liquid water. Faisal et al. [36] investigate and understand weather conditions related to aircraft icing to improve ice prediction. Mei et al. [37] studied the application of the High-Resolution Rapid Refresh model in ice accretion prediction and improved it. Bowyer et al. [38] used satellite data to infer the icing potential in Europe, Asia, and Australia and compared it with the measured results. Many scholars have also researched the prediction of the type of condensate [39–41], hoping to find out the distribution range of particles that are easy to cause aircraft icing, such as SLD [22], to give early warning to pilots. Reisner et al. [42] studied the cloud microphysical characteristics in detail, which played an important role in weather modification and aircraft icing environment research. The National Transportation Safety Board (NTSB) has issued a large number of early warning reports on aircraft icing [43]. The premise of forecasting icing is that we need to have a clear judgment of the general distribution environment of aircraft icing.

China has a vast territory, diverse climates, and large differences in temperature and humidity in time and space. Therefore, research on the distribution of areas prone to icing of Chinese aircraft is very complicated. Wang et al. [44] analyzed the climatic characteristics of aircraft icing accretion in the most recent 40 years using the power spectrum. Yang [45] used reanalysis data to study the validity and accuracy of ice accretion prediction. Wang [46] summarized the distribution characteristics of Chinese aircraft icing routes and established a prediction model of Chinese aircraft ice accumulation using multiple linear regression and neural networks.

Huang constructed the climatic region of aircraft icing in China after a statistical analysis of high-altitude climate data, as shown in Figure 1 [47]. Area I is the most prone to icing and includes most of the northeast and the Qinghai-Tibet Plateau. Area II is more prone to icing and extends from the East Coast to the central region of the western region. It is difficult to accumulate ice in Area III, which is located mainly in the south of the middle and lower reaches of the Yangtze River. Area IV is the area where it is most difficult to accumulate ice and includes the main regions of North China, Xinjiang Province, and Liaoning Province.

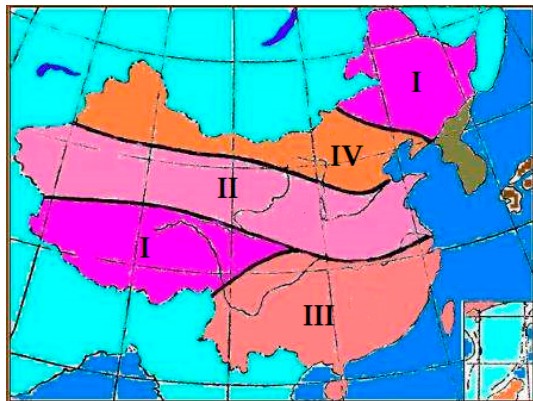

**Figure 1.** Climatic region of aircraft icing in China.

However, the temperature and humidity in the different seasons of China's different regions show great variations. Before and after the rainfall process and the snowfall process, the liquid water in the air may be significantly reduced and may have an impact on the aircraft icing environment. Therefore, based on Figure 1, this paper discusses the changes in the ice index intensity and its main distribution altitude in different seasons in different regions, including the changes of the icing index at middle and low altitude before and after large-scale rainfall and snowfall.

## 2. Materials and Methods

### 2.1. Materials

The data used in this paper include FY-2 Satellite data and sound data.

The satellite data are collected from the multichannel spin scan radiometer data of the FY-2 Satellite. The visible infrared spin scanning radiometer (VISSR) is one of the main payloads of the FY-2 Satellite, and every half hour, a panoramic original cloud image covering 1/3 of the global area can be obtained. VISSR has three channels of visible light, infrared, and water vapor. With three channels of data, various satellite meteorological products can be further obtained. The data used in this paper include the snow cover distribution of the FY-2E satellite and the precipitation estimation of the FY-2F satellite. Satellite data was downloaded on China Meteorological Data Service Center (CMDC).

The sounding data are taken from the Department of Atmospheric Science, University of Wyoming (UW), which supplies daily 0 h and 12 h data. The sounding data website is http://www.weather.uwyo.edu/upperair/sounding.html. The physical quantities used in this paper include altitude, temperature, and RH. The second component uses the data of the current month, and the third component uses the data of 0 h of the current day. It should be noted that, in Section 2, the website is missing four days (on 1, 2, 6, and 7 January) of 0 h data and one day (on 24 April) of 12 h data.

### 2.2. Icing Index

The International Civil Aviation Organization (ICAO) recommends using the icing index 'I$_C$' [48] to describe the icing environment:

$$I_C = 2(RH - 50) \cdot \frac{T(T + 14)}{-49} \tag{1}$$

In Equation (1), 'RH' is the relative humidity (%) and 'T' is the temperature (°C). When RH is lower than 50% or T is not in the range of −14 °C to 0 °C, it is considered that it is impossible to freeze. The I$_C$ output range is from 0–100, and the larger the value, the stronger the icing possibility. When the temperature is −7 °C and RH reaches 100%, the maximum value of I$_C$ is 100. The strength criterion

of $I_C$ is given as follows: if $0 \leq I_C < 50$, slight icing occurs; if $50 \leq I_C < 80$, moderate icing occurs; if $I_C \geq 80$, serious icing occurs.

### 2.3. Interpolation Method and Software Implementation

The interpolation method is a basic method in numerical analysis and is used to supply a continuous function for discrete sample points. This continuous function passes through all existing sample points, i.e., for existing sample points: $(x_1, y_1), (x_2, y_2), \ldots, (x_N, y_N)$, the constructor $P(x)$ satisfies:

$$P(x_i) = y_i, i = 1, 2, \ldots, N \tag{2}$$

Thus, $P(x)$ is known as the interpolation function. According to the uniqueness theorem, if the interpolation function is a polynomial function, the interpolation result satisfying Equation (2) is unique. The common interpolation methods include Lagrange interpolation, Newton interpolation, Hermite interpolation, cubic spline interpolation, and others. Considering that the credibility of the region interpolation outside the sample is low, all of the conclusions in this paper are discussed in the interpolation region within the sample range.

Software MATLAB R2015b contains many interpolation functions, which can be calculated according to demand. Function 'interp1' can interpolate one-dimensional data, and function "scatteredinterpolant" can interpolate two-dimensional or three-dimensional data. In this paper, these two functions are used to interpolate data. In Section 3, 'interp1' is used to interpolate daily data along the altitude, and the interpolation points are every hundred-meter to 15 km above the station altitude. In Section 4, "interp1" is used to interpolate the data of each station to each kilometer between 1 km and 6 km. At the same altitude, "scatteredinterpolant" is used to interpolate to the entire area, forming grid point data with a resolution of $0.1° \times 0.1°$. The interpolation method in this paper is linear interpolation. The "m_map" package is used in plotting.

## 3. Seasonal Distribution of Icing Index

According to the four regions in Figure 1 that are divided by the difficulty of the aircraft icing, each region selects a near-ocean site and an inland site, as shown in Figure 2. The eight stations are Changchun station and Lhasa station in Area I, Zhengzhou station and Kashgar station in Area II, Shanghai station and Kunming Station in Area III, and Beijing station and Altay station in Area IV. The selected lengths of time are January, April, July, and October 2019, which represent winter, spring, summer, and autumn, respectively. The temperature and RH are interpolated to obtain the icing index, and the results are shown in Figures 3–10. The "station" in Figures 3–10 means the altitude of the station.

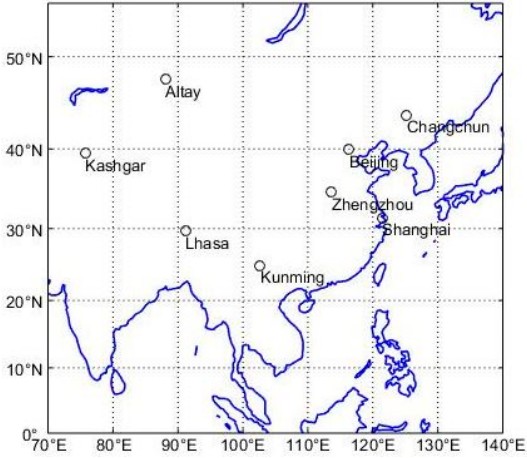

**Figure 2.** Coordinates of 8 sounding stations.

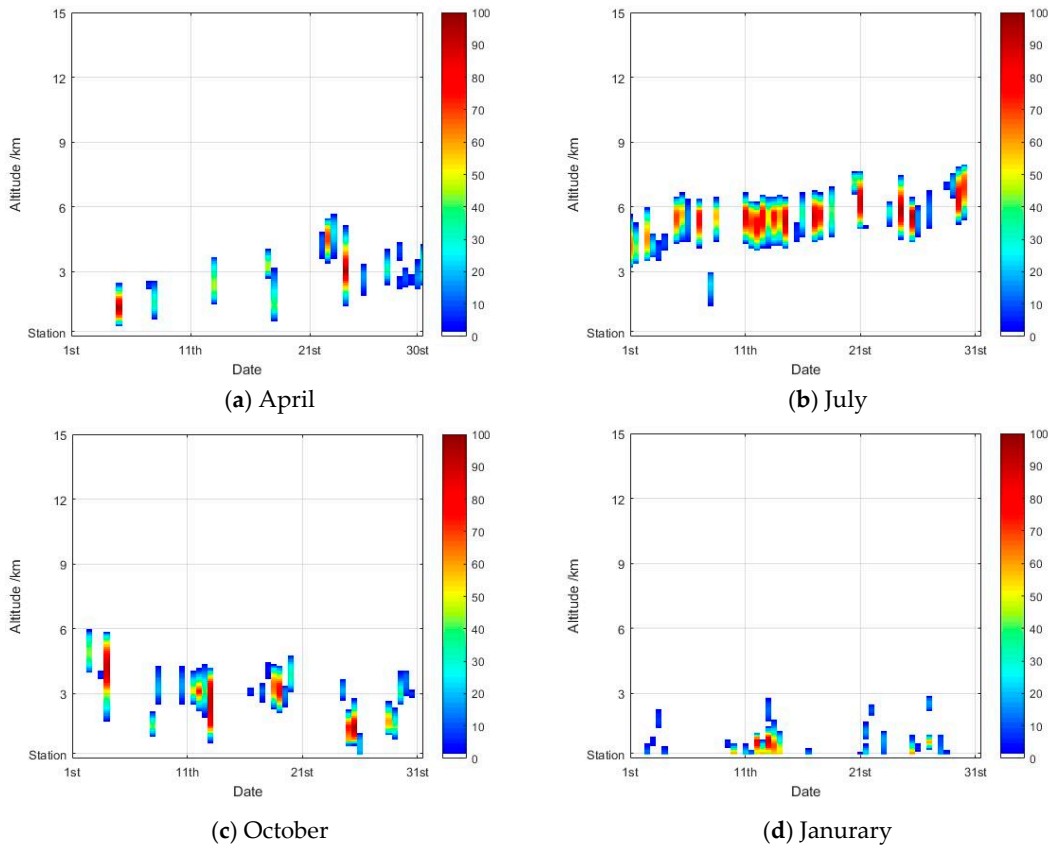

**Figure 3.** Icing index of Changchun Station (Area I). (**a**) April, (**b**). July, (**c**). October, (**d**). Janurary.

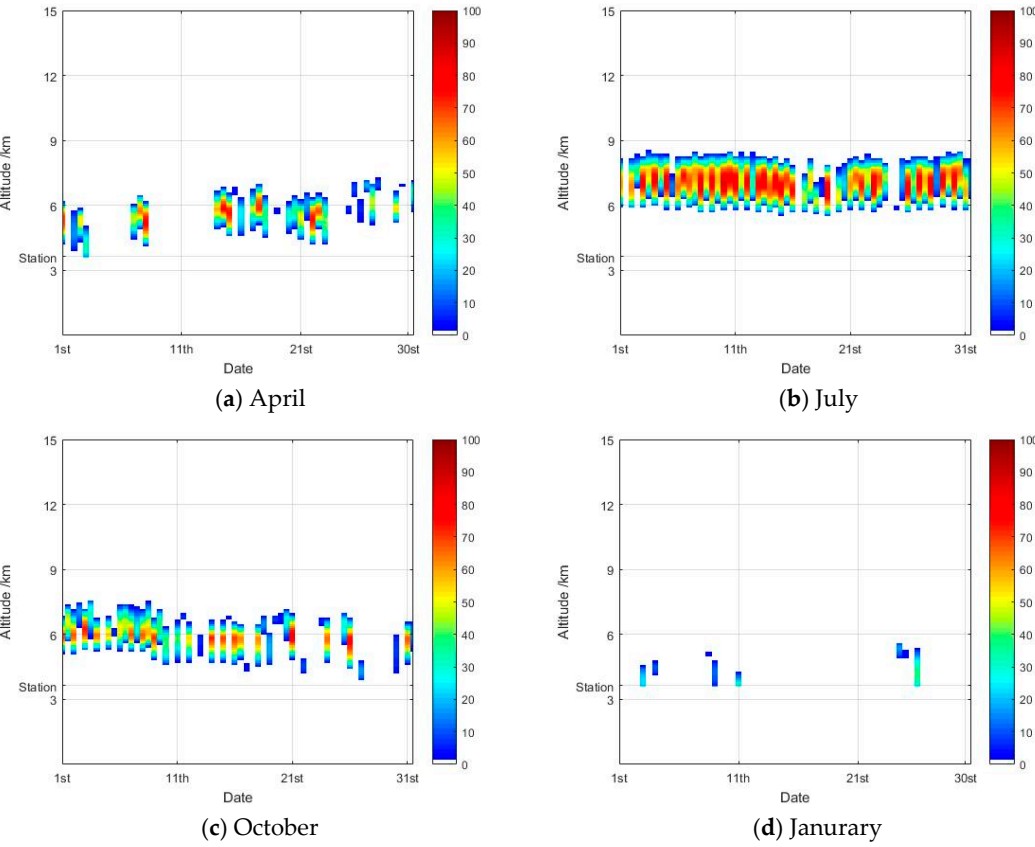

**Figure 4.** Icing index of Lhasa Station (Area I). (**a**) April, (**b**). July, (**c**). October, (**d**). Janurary.

As can be seen in Figure 3, the distribution of the icing index at Changchun Station shows that there is an icing area for about 1/3 in spring, which is below 6 km with mild icing mainly. In summer, icing has the longest time and is highly stable in the range of 4 km to 7 km. The duration of icing in autumn accounts for about 1/2, located below 6 km, mainly mild icing, but also severe icing for several days, basically similar to spring. In winter, icing is below 3 km, which gradually weakened from near the surface to low altitude.

As shown in Figure 4, the icing distribution at Lhasa station shows that the icing time accounts for approximately 1/2 of spring and that the icing altitude above and below 6 km consists mainly of light icing, with a small amount of moderate icing. In summer, daily icing occurs, which is stable between 6 km and 8 km, and the intensity is a large amount of moderate icing and even serious icing. The icing deposition in autumn is similar to that in spring, but it lasts longer. In winter, icing is notably weak and rarely occurs.

The icing index of Zhengzhou station is presented in Figure 5. The duration of icing at Zhengzhou station is approximately 1/3, and the altitude is between 3 km and 6 km, mainly light icing with a small amount of moderate icing. During summer, the duration of icing is approximately 2/3, and the altitude of icing is between 5 km and 7 km. The distribution of icing in autumn is similar to that in spring but is more continuous. The most serious icing occurs in winter, is located from the ground to 5 km, and is mainly moderate and severe.

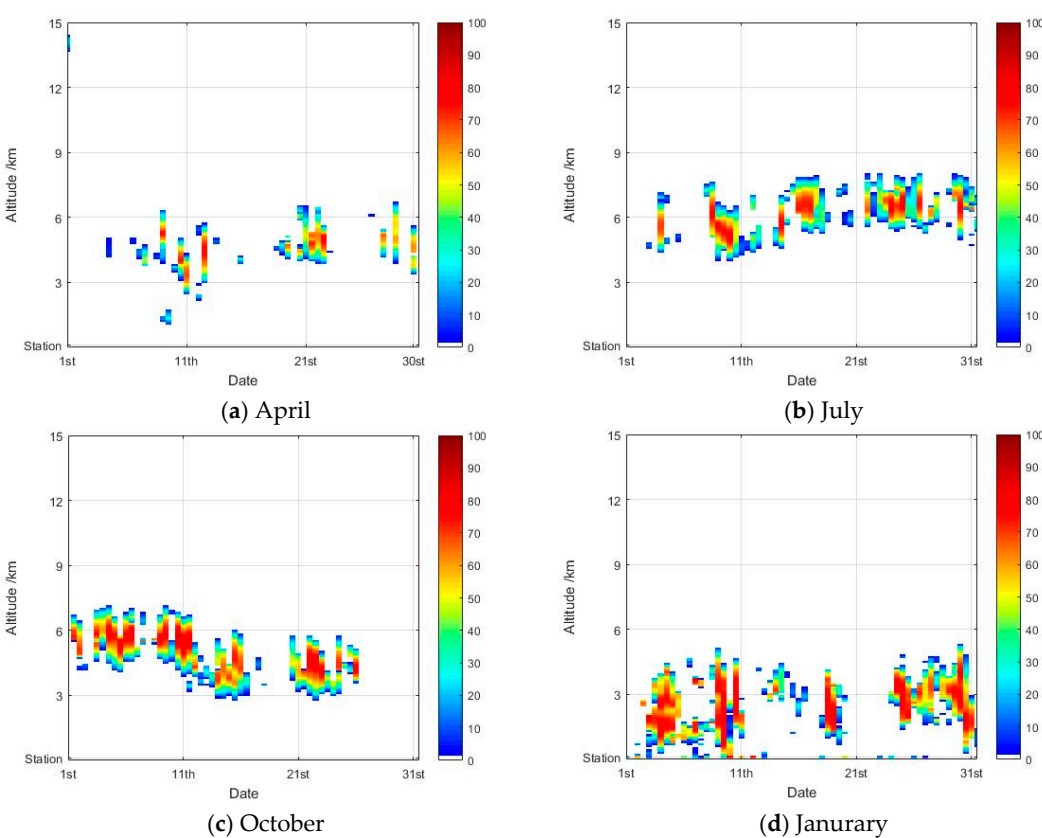

**Figure 5.** Icing index of Zhengzhou Station (Area II). (**a**) April, (**b**). July, (**c**). October, (**d**). Janurary.

The icing index of Kashgar station is presented in Figure 6. Icing at Kashgar station occurs in spring, at altitude primarily between 4 km and 6 km, with mainly light icing and little moderate icing. In summer, icing occurs with an altitude between 5 km and 7 km and is mainly light icing and a certain amount of moderate icing. The icing distribution in autumn is similar to that in spring, and the duration is almost 2/3. In winter, icing occurs from the ground to 3 km and is mainly light icing and selected moderate icing, with little icing above 3 km with a weak icing intensity.

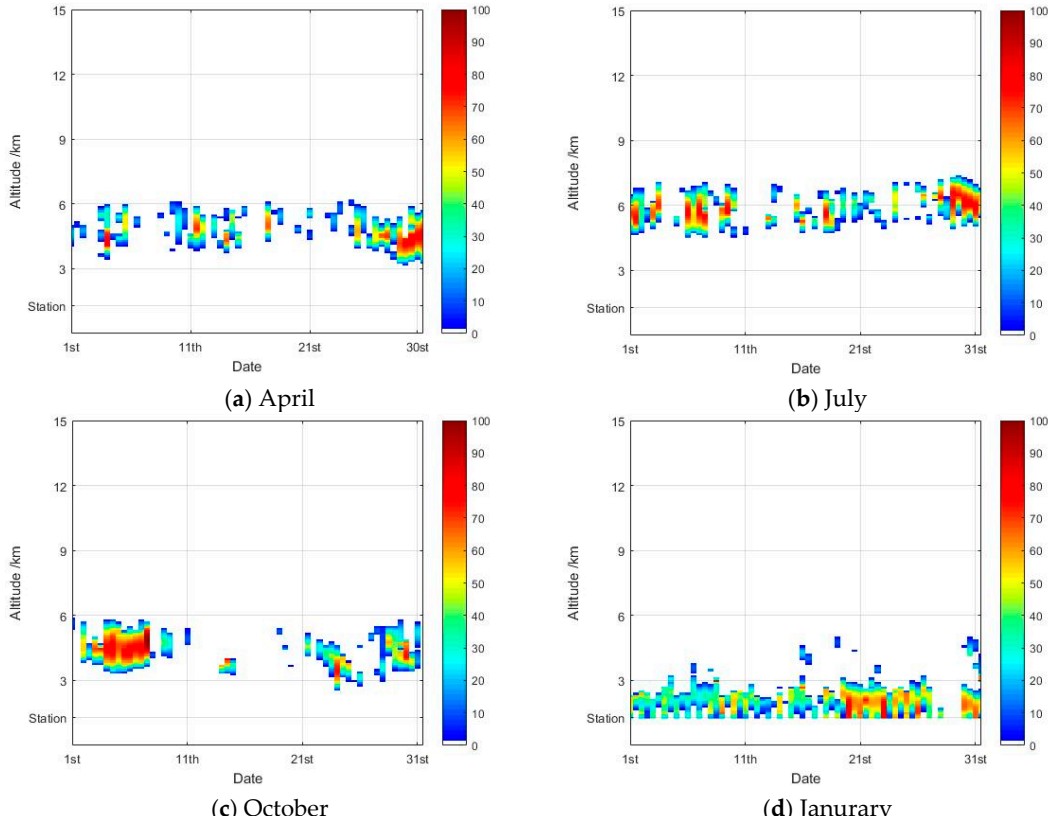

**Figure 6.** Icing index of Kashgar Station (Area II). (**a**) April, (**b**). July, (**c**). October, (**d**). Janurary.

As shown in Figure 7, in spring, the icing distribution of Shanghai station is relatively discrete, with an altitude between 3 km and 7 km, and is mainly light and moderate icing and a small amount of serious icing. The duration icing accounts for almost 1/2 of summer, with an altitude between 5 km and 8 km, and is mainly light and moderate icing. The distribution of icing in autumn is similar to that in spring. The icing in winter is the most serious, from the ground to 6 km, and is mainly medium and grievous icing.

As shown in Figure 8, the duration of icing at Kunming station is longer than 2/3 of spring, with an altitude of 5 km to 6 km, and is mainly light icing and a small amount of moderate icing. The icing in summer is generally continuous, with an altitude of 6 km to 8 km, and is mainly moderate icing. The duration of icing is approximately 2/3 of autumn, the altitude is stable at 5 km to 7 km, and the icing intensity is mainly light with little moderate. The duration of icing is relatively short, and the altitude is maintained at a low altitude of 3 km to 4 km in winter.

As displayed in Figure 9, the icing at Beijing station lasts for approximately 1/2 of spring, and the altitude changes greatly, but the range is generally below 6 km. The intensity of icing is moderate or serious most of the time. In summer, the icing altitude is between 4 km and 8 km, and the intensity is mainly light icing. The distribution of icing in autumn and spring is nearly identical and different from other seasons, there is almost no icing in winter.

The icing index distribution of Altay station (Figure 10) shows that the icing time in spring accounts for approximately 1/2, the altitude range is generally below 5 km, and mainly there is mild icing and a small amount of moderate icing. The icing time in summer accounts for approximately 1/3, with an altitude between 4 km and 6 km, and light icing is the main component. The icing takes about 1/3 of the time in autumn, and the altitude is widely distributed, and icing may also occur near the ground, with a small amount of moderate icing. In winter, icing occurs, extending from the surface to 3 km.

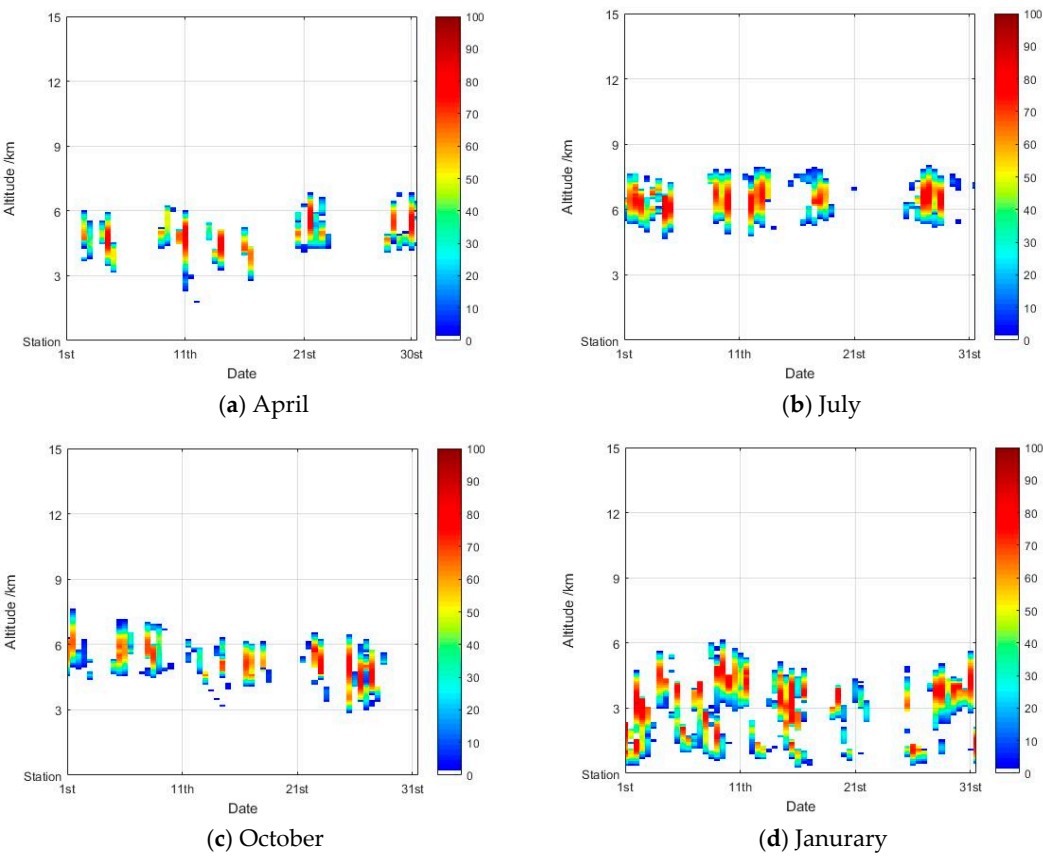

**Figure 7.** Icing index of Shanghai Station (Area III). (**a**) April, (**b**). July, (**c**). October, (**d**). Janurary.

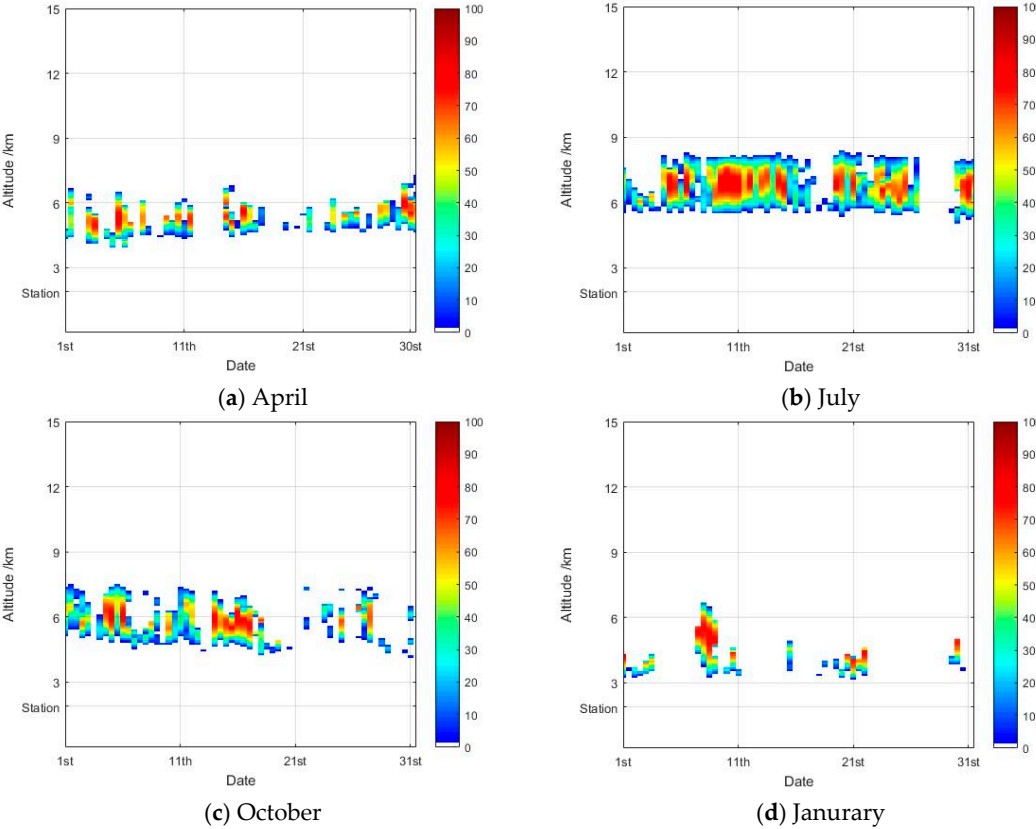

**Figure 8.** Icing index of Kunming Station (Area III). (**a**) April, (**b**). July, (**c**). October, (**d**). Janurary.

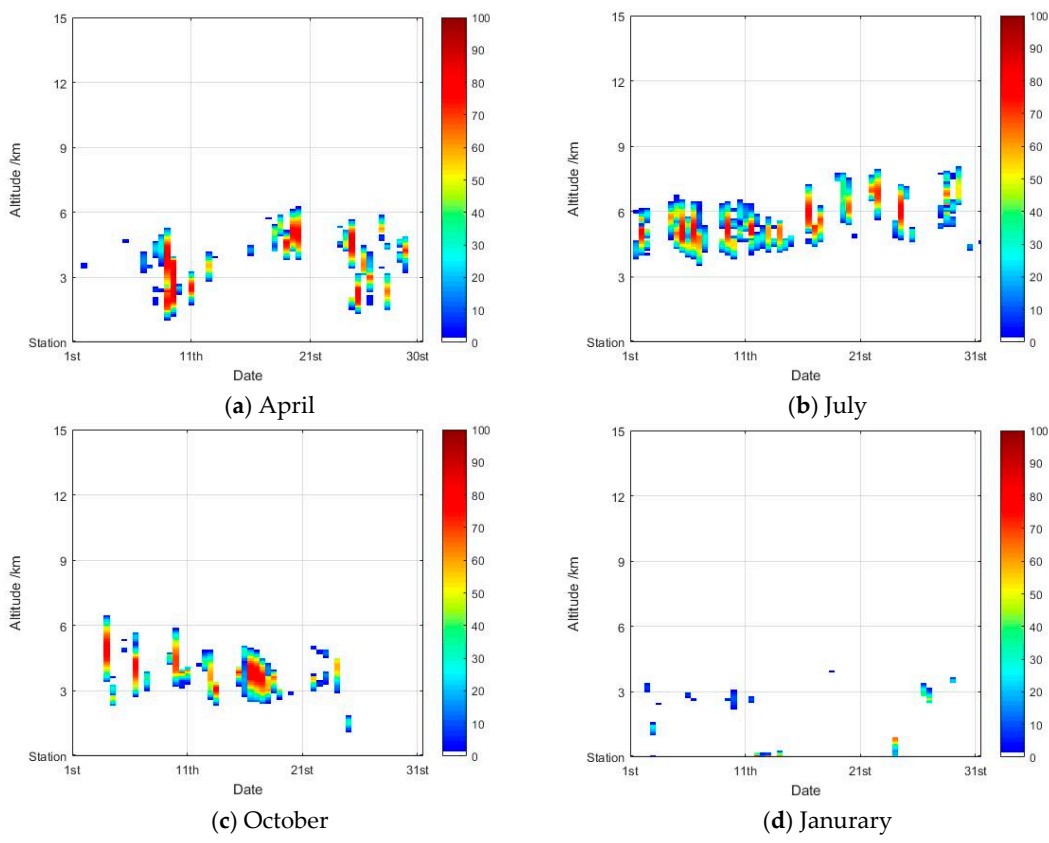

**Figure 9.** Icing index of Beijing Station (Area IV). (**a**) April, (**b**). July, (**c**). October, (**d**). Janurary.

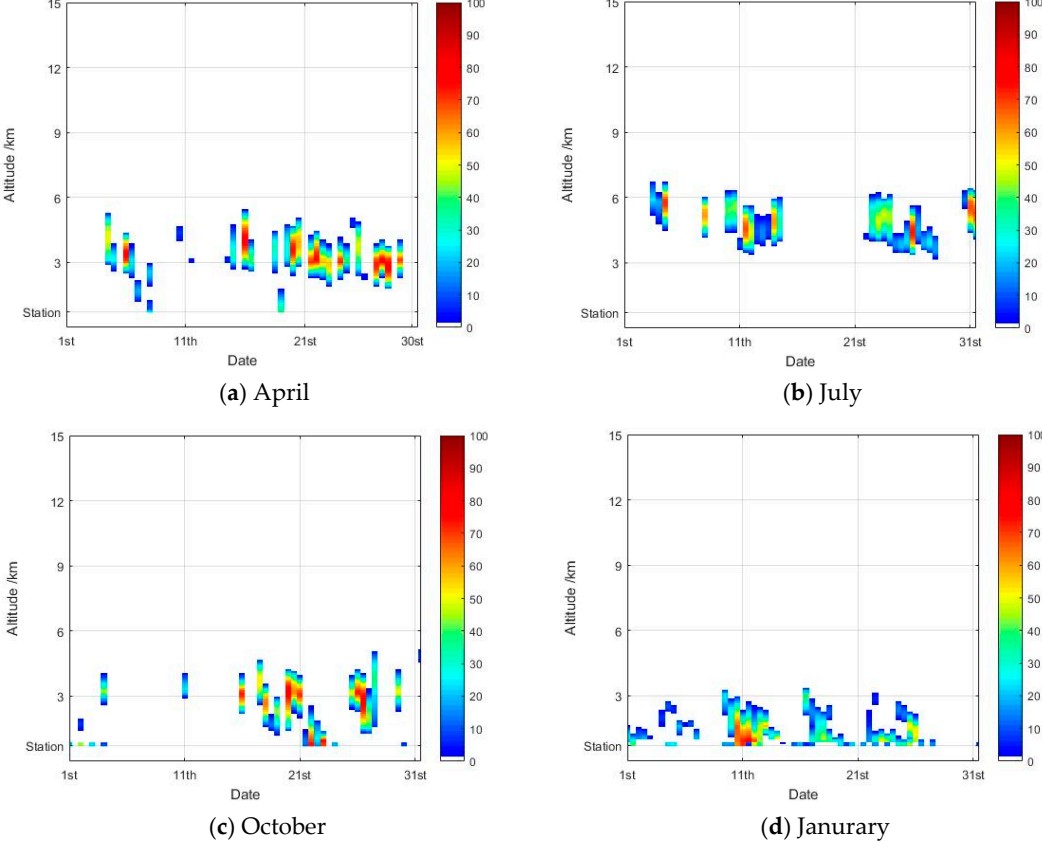

**Figure 10.** Icing index of Altay Station (Area IV). (**a**) April, (**b**). July, (**c**). October, (**d**). Janurary.

## 4. Impact of Rainfall and Snowfall on Icing Index

When certain weather processes occur, the temperature and humidity environment change significantly in a short period of time, which is difficult to clearly show in monthly scale statistics and must be discussed separately. Based on the conclusion of the second section, icing occurs primarily in the near-ocean region, in Area I when in summer, in the Area II and Area III when in winter. In this section, we analyze the influence of two different weather processes on the icing index of different altitude before and after the summer rainfall in Northeast China and the winter snowfall in East China. It should be noted that the results for 1 km, 2 km, and 3 km are omitted in Section 4.1 and those for 6 km are omitted in Section 4.2 because there is no icing in the entire area at the above altitude.

### 4.1. Rainfall Process

In August 2019, high-intensity rainfall occurred frequently in Northeast China, with the number of precipitation days generally ranging from 12 to 20 days, and in certain special areas, the precipitation days exceeded 20 days. The cumulative precipitation in Heilongjiang Province and Jilin Province was the highest since 1961, and a total of 198 stations in Northeast China experienced extreme continuous precipitation events, which led to the water level rise in Songhua River, Nenjiang River and other sections of the river, and certain areas suffered from rainstorms and floods. Figure 11 shows the precipitation estimation from FY-2F on 11 and 19 August 2019. It can be observed from the figure that rainfall occurred in the entire region of Northeast China (120° E–140° E, 38° N–50° N) on 11 August, and the rainfall mostly ended on 19 August.

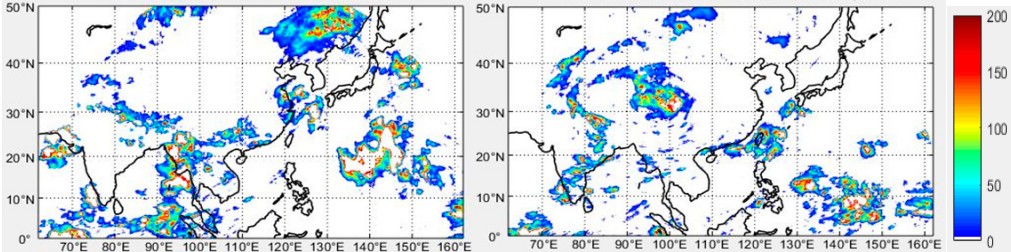

**Figure 11.** Precipitation estimation data based on FY-2F (The left figure is 11 August, and the right figure is 19 August).

We selected 8 sounding stations, such as Harbin station in Northeast China, the station number is shown in Figure 12, and the icing index of different altitude is shown in Figure 13.

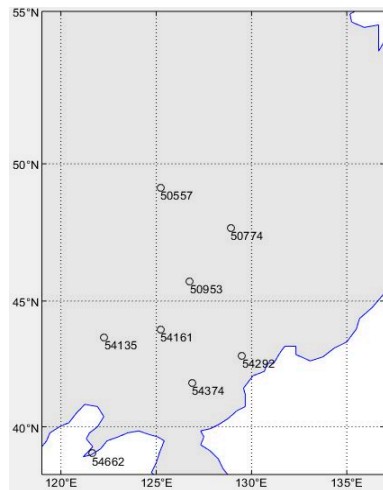

**Figure 12.** Coordinates of 8 sounding stations in Northeast China.

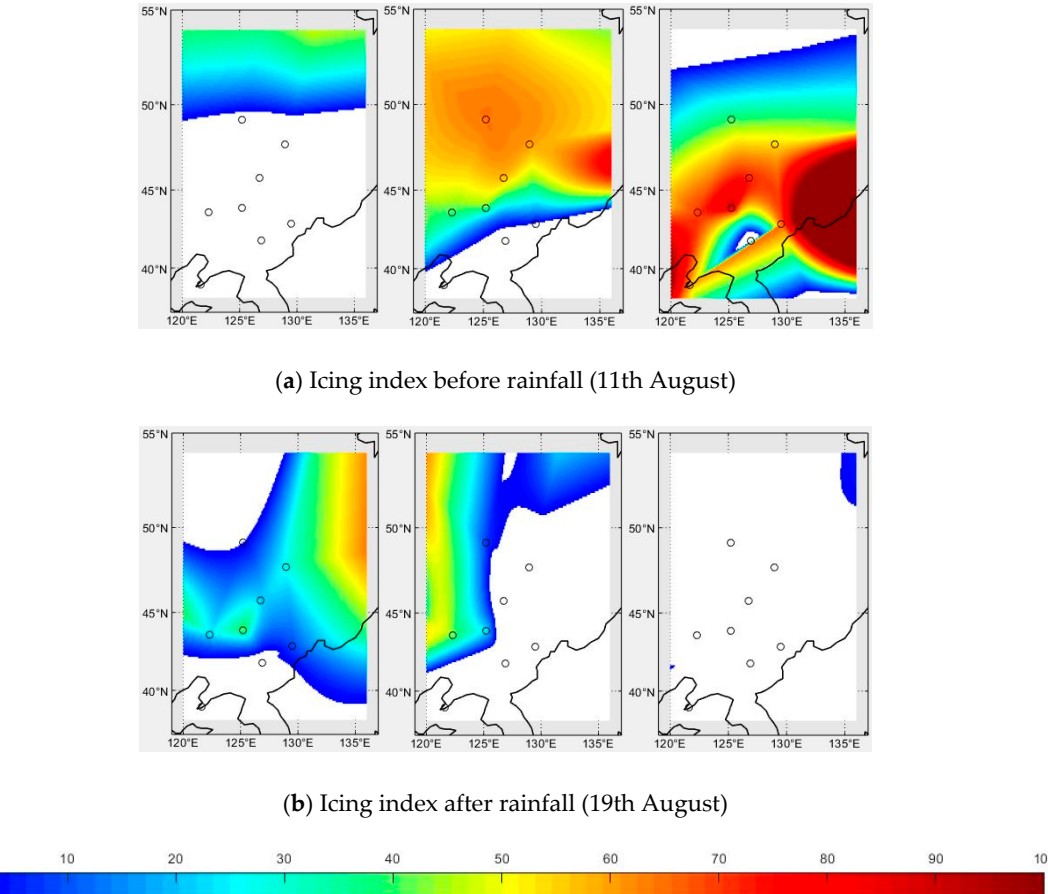

(**a**) Icing index before rainfall (11th August)

(**b**) Icing index after rainfall (19th August)

**Figure 13.** Icing index before and after rainfall (In (**a**,**b**), the altitude from first to third is 4 km to 6 km, respectively).

Figure 13 shows that there is no possibility of ice accretion at a low altitude below 3 km before and after the rainfall process, whereas for the airspace of 4 km to 6 km, the range of rainfall estimation is notably close to the range of ice accretion distribution, which is also consistent with the results presented in Figure 3. The distribution of the icing index at a 5 km altitude on 11 August (Figure 13a) and the index at a 4 km altitude on 19 August (Figure 13b) are most consistent with the rainfall estimation in Figure 11, and the larger rainfall estimation also corresponds to the stronger icing. A possible reason for this result is that a certain proportion of super-cooled large water droplets occur in the rainfall area. The above analysis shows that before the summer rainfall process in Northeast China, ice accretion easily occurs at the altitude of 4–6 km, and the intensity of icing is roughly related to precipitation estimation.

*4.2. Snowfall Process*

In January 2018, three large-scale rain and snow events occurred in China's central and eastern regions (3rd–4th, 5th–7th and 24th–28th), of which the 24th–28th event had the most extensive, long-term and severe impact process during this winter. The accumulated snowfall in the southwest of the Yellow River, the Huaihe River, and the north of regions south of the Yangtze River ranged from 10 mm to 25 mm. The accumulated snowfall reached more than 25 mm, such as in the northeast of the Hunan Province, the north and east of Hubei Province, the central and south of Anhui Province, the southwest of Jiangsu Province, and the north of Zhejiang Province. The snow distribution data from FY-2E on 28 January and 4 February 2018, are shown in Figure 14. The figure shows that the central and eastern regions of China (110° E–123° E, 25° N–40° N) are generally covered with snow

on 28 January, among which Shandong Province, most of Jiangsu Province and the eastern portion of Anhui Province have the thickest snow. By 4 February, the snow in this region has mostly melted, with less snow on the ground.

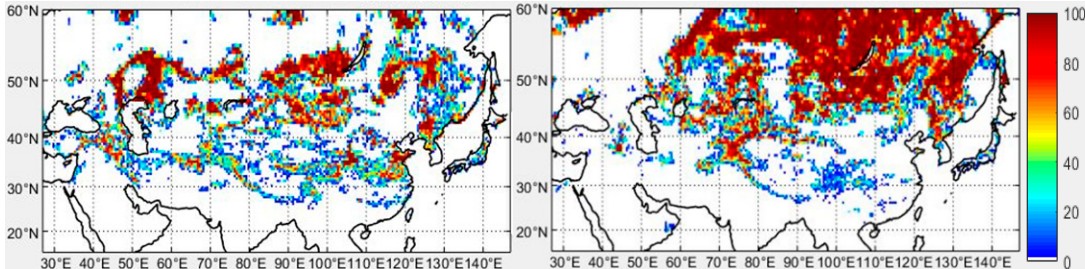

**Figure 14.** Snow cover distribution data based on FY-2E (The left figure is 28 January, and the right figure is 4 February).

As shown in Figure 16, we obtained the icing index of different altitudes by selecting 21 sounding stations such as Beijing station in the eastern portion of China, and the station number is shown in Figure 15.

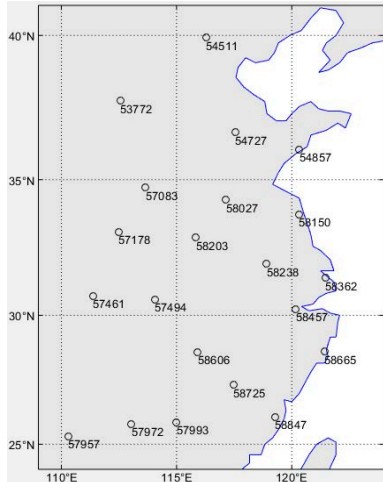

**Figure 15.** Coordinates of 21 sounding stations in the eastern portion of China.

We can conclude that ice accretion is nonexistent in the higher airspace over 5 km, but for the lower airspace below 4 km, when snow is on the ground, an obvious ice accretion area exists. On 28 January (Figure 16a), from 4 km down to 1 km, the ice accretion increases, and in Area II to the north of the Yangtze River, serious ice accretion occurs, which is also consistent with the conclusions in Figures 5 and 7. At the same time, the low-level ice area is similar to the snow area in Figure 14. With the increase in altitude, the difference between the ice area and the snow area gradually increases, and on 4 February, after the snow melts (Figure 16b), the risk of ice accretion at low altitudes is reduced, and only a small amount of area has light ice accretion. The above analysis shows that the winter snow in the east region of China easily causes low-level ice accretion.

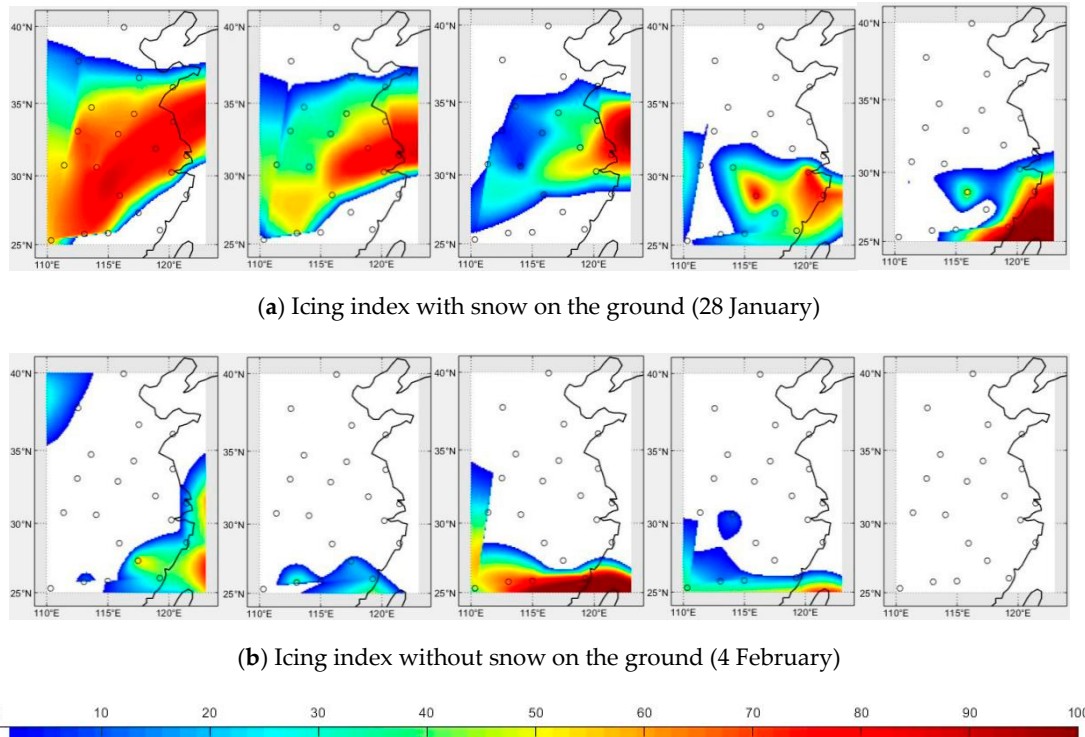

(**a**) Icing index with snow on the ground (28 January)

(**b**) Icing index without snow on the ground (4 February)

**Figure 16.** Icing index with and without snow on the ground (In (**a**,**b**), the altitude from first to fifth is 1 km to 5 km, respectively).

## 5. Discussion

### 5.1. Icing Index Distribution

After comparing the results of the eight sites in Section 3, the main results are as follows:

(1) In general, according to the distribution of icing in the four seasons, the threat of icing in spring is the weakest, mostly distributed almost 3–6 km, with slight icing mainly. The height of icing in summer is increased and relatively stable. The situation is similar in autumn and spring. The icing altitude in winter is mostly close to the ground, and severe icing may occur in Southeast China.

(2) For the same icing climatic region in Figure 1, the icing altitude range and intensity at near-ocean stations are larger and stronger than those at inland stations. A possible reason for this result is that a large amount of liquid water might be present in the low and hollow altitude due to the high humidity in the near-ocean area. At the same time, the existence of the sea-land breeze and the influence of multi-scale weather processes make the duration of icing unstable. In relative terms, the higher altitude of stations in the inland corresponds to a more stable temperature and LWC environment.

(3) For the same site, great differences occur in the distribution of icing in different seasons. The reason for this result is that most areas in East China are affected by the monsoon. Therefore, the moisture content contained in the monsoon inevitably leads to different LWC. In addition, China has a vast territory, complex terrain, and temperature difference between winter and summer is large in most areas. Different temperatures and vapor environments in the four seasons create various icing distributions.

### 5.2. The Impact of Weather Processes

In the process of rainfall and snowfall, we selected two stations with serious ice accumulation: Nenjiang Station (No. 50557) and Nanjing Station (No. 58238) to analyze the main influencing factors of ice accumulation.

As can be seen in Figure 17a, the temperature did not change significantly before and after the rainfall process, but at 4–5 km altitude, RH decreased from near saturation to nearly 50%. As can be seen from Figure 17b, compared during and after snowfall, the air temperature below 2 km dropped about 5 °C, and the temperature drop at 2–5 km was greater, and RH fell from near saturation to less than 30%. Other stations with the possibility of icing accumulation have similar characteristics.

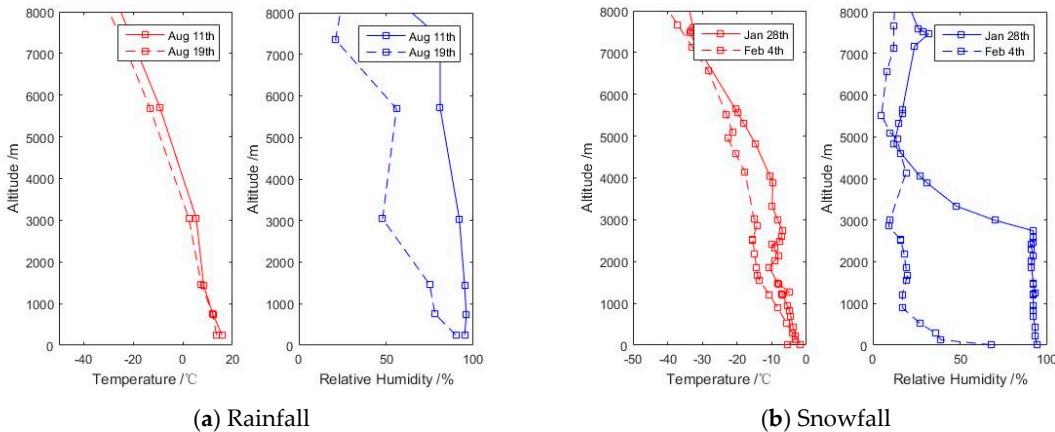

(**a**) Rainfall                                                                    (**b**) Snowfall

**Figure 17.** Height profile of temperature and relative humidity (RH). (**a**) Rainfall, (**b**) Snowfall.

It can be seen that in the large-scale summer rainfall in Northeast China and winter snowfall in East China:

(1) During the summer rainfall in Northeast China, the temperature is relatively high and there is no change before and after the rainfall. However, the obvious drop in RH reduces LWC in the air, causes the possibility of icing is weakened or even disappeared. The data volume in this area has a small number of altitude grid points, so it is difficult to judge the lifting condensation level based on the inversion layer, to accurately judge the cloud height. However, combined with the previous analysis, because the icing area is similar to the rainfall estimation area calculated by satellite data, it is possible that the icing here is related to the cloud, but this conclusion needs further verification.

(2) The change of temperature and humidity in the snowfall process in East China in winter has obvious cold front transit characteristics. The confluence of warm and cold flow in front of the cold front caused snowfall. The ground temperature was slightly below 0 °C, the humidity was high, and there was a lot of liquid water at low altitude, which was likely to cause the aircraft to freeze. When the snow has melted, the cold front has completely passed, and the dry and cold air occupies the height of the original warm and humid air. Therefore, there has been a significant cooling, and the humidity has dropped significantly. The liquid water in the air has been consumed and the aircraft is not easy to freeze.

(3) After the process of rainfall and snowfall, the temperature and RH decreased. A decrease in RH will inevitably cause a decrease in the icing index, and even icing may not occur. However, a decrease in temperature does not necessarily bring about a decrease in icing index: (1) if it just drops below 0 °C, it may be more conducive to the appearance of super-cooled water and increase LWC, to increase icing possibility. (2) If the temperature drops below −20 °C or even lower, LWC will decrease and icing possibility will decrease. (3) The temperature drops stably at each level will bring the 0 °C level downward, which will reduce the altitude of the icing area.

## 6. Conclusions

Based on the climatic region of aircraft icing in China and the icing index, we investigated data from sounding sites in different icing zones in different seasons using linear interpolation and analyzed the impact of summer rainfall in the Northeast China and winter snowfall in East China on the icing index on 1–6 km altitude. The conclusions are given as follows:

(1) The distribution of icing varies greatly in different regions, seasons, and altitude. This phenomenon comes from differences in temperature and cloud microphysical characteristics such as LWC and MVD by different environments. The icing index calculated by temperature and relative humidity can effectively reflect this.

(2) Before the summer rainfall in Northeast China, icing is prone to occur at altitudes of 4 km to 6 km, and the intensity of icing may be related to precipitation estimation. Snow on the ground in winter in East China is likely to cause low-altitude icing and its intensity may be serious. However, when the rainfall and snowfall process is over, the LWC in the air is decreased, and the threat of icing is significantly reduced. This feature is helpful for early warning of icing for the take-off and landing of transport aircraft and the flight of general aviation.

(3) Ideas for further research: the above conclusions can qualitative analysis and a theoretical basis for the prediction of aircraft icing and improvement in flight safety. Considering the limitations of the interpolation method, additional station information and detailed meteorological data are required for further verification of the conclusion of this paper, as well as actual flight testing, if necessary.

**Author Contributions:** B.X. conceived and designed the experiments, performed the experiments, analyzed the data, and wrote the paper; J.W. and J.C. helped in the discussion and revision; J.W. reviewed the manuscript. All authors have read and agreed to the published version of the manuscript.

**Funding:** This research was funded by National Natural Science Foundation of China (41905026); the Natural Science Foundation of Jiangsu Province (BK20170945); the 63rd Batch of China Postdoctoral Science Foundation in General (2018M631554); Open Fund by Key Laboratory of Aerosol-Cloud-Precipitation of CMA-NUIST (KDW1703); Key Laboratory of Middle Atmosphere and Global Environment Observation (LAGEO-2019-05).

**Acknowledgments:** The author B.X. is particularly grateful to the questions from the defense group in the opening report of the master's thesis, which reminded me of the need to study the icing altitude distribution in detail. The authors would like to acknowledge the CMDC and the UW for providing the observation data and precise products.

**Conflicts of Interest:** The authors declare no conflict of interest.

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
