# Peer review of "The Distribution of Aircraft Icing Accretion in China—Preliminary Study"

_atmosphere, doi:10.3390/atmos11080876_

Round 1
Reviewer 1 Report
Dear Authors,
Thank you for your contribution. The paper touches one of the most critical issues in aviation safety. Icing and ice accretion are particularly dangerous for smaller aeroplanes which are not able to avoid adverse weather.
With your paper, you presented a considerable amount of data concerning, practically, the whole territory of China, which brings an excellent overall picture of icing and ice threat across the country.
However, before I pass your paper, I would dare to point out several remarks, which may improve the scientific soundness of the publication. I would appreciate if you could address the following issues:
- The title of the paper: The Distribution of the Icing Index and the Influence of Rainfall and Snowfall on the Icing Environment. Compared to the factual content of the article, the proposed title is too broad by far. Although you developed, to some extent, all the keywords from the title, the centre of gravity is somewhere else, unfortunately. Most of the paper is consumed by the presentation of the results of data postprocessing using two formulas and extrapolation function. The influence of the different types of precipitation is treated in a quantitative manner without a deeper development.
- The introduction part requires further development. There is not enough information to set the scene for the presented research. The reader knows the current approach in China. However, China is not an isolated island and would be useful to mention how it is done in different parts of the world - better, worse, differently? This would also allow Authors to justify their current approach.
- The research method looks quite simplistic. The authors based their analysis on two formulas and algorithm in Matlab. Then, they processed the data from sounding stations collected across the year. Finally, a comparative (I would say qualitative) analysis of lateral profiles with satellite images was performed.
I would consider this as insufficient for scientific research paper. In the current form, the paper presents an observation report with post-processed data.
There is little information about the span of data: presumably, it was one year only. Therefore, how representative was this particular year compared to the longterm observations (5-10 years)?
The paper contains a vast number of charts (Figures 3 - 10) with short comments after each. Going back to my remark about the title, it is not necessary to present all of them in the main body, unless we focus on the distribution of the icing index only. The results from one station are representative of the method, and the rest may go to the appendix. The conclusions following the charts (lines 228 - 274) look superficial and sound apparent sometimes. I would expect a more in-depth comparative analysis with the results from previous years, results of other researchers, models, results from flight tests, or information from PIREP or AIRMET reports. This part of the research lacks verification part completely.
The second part of the paper, the Influence of Rainfall and Snowfall on the Icing Environment, limits the analysis to the comparative analysis of the lateral profile of the data with satellite images on certain days in the year. The conclusions drawn are descriptive only, with no more profound analysis, similarly to the previous part.
To conclude the remarks about the core of the paper, I must say that the title does not reflect the content entirely. In the present form, the appropriate title may refer to "icing areas over China - preliminary study".
Unfortunately, I reckon the paper needs a major revision at this point.
Moreover, I found several minor issues listed below:
- Line 116 - How can you say "T is in the range of -14 ℃ to 0 ℃, it is considered that there is no ice accretion", if earlier on (lines 39 - 40) you said the icon is most likely in "with an external temperature generally between -15℃ and -3℃". Obviously, the latter sounds reasonable, and I hope there is a typo in formula (1) which had not affected the results.
- Formulas need readjustment - in my copy, they are shifted into a couple of lines.
- Why the vertical scale on charts (Figure 3 and the following ones) is up to 15 kilometres if the maximum altitude of icing is 9 kilometres only?
- Please, be careful using height and altitude. Now it is inconsistent.
- Figure 1 needs improvement. Now it looks like a poor scan from a photocopy.
- Figure 16 - what do you mean by: "the distance from first to fifth is 1 km to 5 km, respectively". What distance?
I hope my comments would help you improve your paper.
Good luck
Your reviewer
Reviewer 2 Report
Please also see attached PDF.
July 06 2020
Decision: major revisions
Title: The Distribution of the Icing Index….
by Jinhu Wang 1,2,3,4*, Binze Xie1, Jiahan Cai1
General: This paper focuses on icing index climatology over China and used RH, T, and satellite obs. There are serious issues and statements done on based on RH and T obtained from radiosonde observations. Goal of the paper is not clearly stated and method is very weakly explained, specifically how obs are used. For example, they talk about satellite obs but not retrievals or products.
My comments are given in the attached pdf. The lines deleted means you take out the sentence or modified it. There are many of them, language needs to be improved.
Results are given for individual stations but not summarized in a table, no need to show for all stations but 2 extreme conditions will be enough.
No discussion is given properly and discussions and conclusion are very short.
Conclusions are not supported and there are many claims that are correct.
Suggested papers:
- Gultepe,I., Agelin-Chaab, J. Komar, G. Elfstrom, F. Boudala, and B. Zhou, 2019: A meteorological supersite for aviation and cold weather applications. Pure and Appl. Geophy. doi.org/10.1007/s00024-018-1880-3. V. 176, No. 5, 1977-2017.
- Gultepe, I., R. Sharman, P.D.Williams, B. Zhou, G. Ellrod, P. Minnis, S. Trier, S. Griffin, S.S. Yum, B. Gharabaghi, W. Feltz, M. Temini, Z. Pu, L.N. Storer, P. Kneringer, M.J. Weston, H.Y. Chuang, L. Thobois, A.P. Dimri, S.J. Dietz, Gutenberg, M.V. Almeida, and F.L. A. Neto, 2019: A Review Of High Impact Weather For Aviation Meteorology . Pure and Applied Geophysics. V. 176, No. 5, 1869-1923.
- Gultepe, I., M. Pavolonis, B. Zhou, R. Ware, R. Rabin, W. Burrows, J. Milbrandt, and L. Garand, 2015: Freezing Fog and Drizzle Observations. SAE International Conference on Aircraft and Engine Icing and Ground Icing, 22-25 June 2015, Prague, Czech Republic. SAE Technical Paper 2015-01-2113, 2015, doi:10.4271/2015-01-2113. 15pp

Round 2
Reviewer 2 Report
This is not written coherently, many sentences are lacking of scientific concepts and thoughts. see my colored areas on the paper but these are not limited, i am sure there are many others.
Icing index equation is given but not explained how it is used, what layer or level? what are the issues using this equation, needs to be discussed in discussions.
Concs are very rough and not clear.
In fact my previous suggestions are not done properly. If this paper is not improved as requested, i will suggest to be rejected.
See my attached files also.
